# A Comparison of Dynamic Hip Screw and Two Cannulated Screws in the Treatment of Undisplaced Intracapsular Neck Fractures—Two-Year Follow-Up of 453 Patients

**DOI:** 10.3390/jcm8101670

**Published:** 2019-10-12

**Authors:** Harald Kurt Widhalm, Richard Arnhold, Hannes Beiglböck, Alexandru Munteanu, Nikolaus Wilhelm Lang, Stefan Hajdu

**Affiliations:** 1Department of Orthopedics and Traumatology, Clinical Division of Traumatology, Medical University of Vienna, Waehringer Guertel 18–20, A–1090 Vienna, Austria; richardarnhold@gmail.com (R.A.); hannes.beiglboeck@hotmail.com (H.B.); nikolaus.lang@meduniwien.ac.at (N.W.L.); stefan.hajdu@meduniwien.ac.at (S.H.); 2Department: Medical School, University College London, London WC1E 6BT, UK; alexandru.munteanu.14@ucl.ac.uk

**Keywords:** osteosynthesis, femoral neck fractures, intracapsular, dynamic hip screw, cannulated screw, avascular necrosis

## Abstract

One of the most common fractures is that of the intracapsular femoral neck; however, the optimal implant for head-preserving treatment remains controversial. The aim of the study was to compare the outcomes of treating undisplaced intracapsular femoral neck fractures with either the dynamic hip screw (DHS) or the double cannulated screw fixation (CSFN). This retrospective cohort study analysed the data of 453 patients, with a mean age of 76.9 years, whose intracapsular fractures were treated with the DHS or CSFN between 2005 and 2013. The analysis focused on the rates of revision surgeries and complications; however, the impact of confounding exogenous factors, such as smoking and alcohol, were also considered. No significant difference was observed between the revision rates of DHS and CSFN (15.0% vs. 13.1%; *p* = 0.565). According to the complication rate, the advantage in favour of the CSFN was not significant (20.5% vs. 13.1%, *p* = 0.038). The use of the DHS was associated with a 13 min longer surgery (*p* < 0.0001) and a one day longer hospitalization (*p* = 0.242). Excessive consumption of alcohol was associated with an increased incidence of avascular necrosis (18.6% vs. 8.7%, *p* = 0.035). The choice of implant showed no significant impact on rates of revision surgery and complications. In terms of socioeconomic factors, the fixation with two cannulated screws was more favourable, making it the more cost-effective and less stressful method.

## 1. Introduction

Due to the demographic development of an increasing elderly population, the treatment of the most common fracture among this age group, the intracapsular femoral neck fracture, is of ever-growing importance. Population forecasts predict an increase in people over 65 such that by the year 2100 they will make up 29.4% of the total Austrian population [1]—a similar trend can be expected throughout Europe and North America.

The frequency of age-associated bone fractures is expected to rise as life expectancy continues to increase [2]. The principle factor responsible for this trend is the age-related bone demineralisation of the proximal femur compounded by the high prevalence of vitamin D deficiency [3]. Furthermore, these changes are compounded by the higher rates of falls among the elderly [4]. Cumulatively, these factors are responsible for approximately 90% of femoral neck fractures [5]. 

Considering the rising need for surgical treatment of intracapsular femoral fractures, as well as the range of osteosynthetic therapies, it is imperative to assess the quality of different care options to ensure optimal treatment. The economic aspect of care is gaining importance as well, favouring cheaper non-inferior interventions.

Today’s most popular head-preserving methods include the dynamic hip screw (DHS) and the percutaneous screw fixation using three cannulated screws. In this retrospective study we examine outcomes of intracapsular hip fractures treated using a DHS (DePuy Synthes Companies, Zuchwil, Switzerland) compared to those treated with double cannulated screw fixation (CSFN) (Sanatmetal Kft, Eger, Hungary), similar to the Manninger screw. The widespread and established multiple cannulated screw fixation uses three screws that have a 7.3 mm diameter, while the CSFN uses two cannulated screws with a diameter of 9.5 mm. Rotational stability in surgical treatment of femoral neck fractures remains an important topic. Therefore, rotational stability of the femoral head can be provided in patients treated with screw-blade fixation by precise centre/centre placement of the lag screw or by an additional anti rotation screw (7.3 mm diameter) cranially to the lag screw, in addition to using the screw-anchor fixation system [6,7].

The aim of the study was to compare DHS, the gold standard procedure, to CSFN in patients with undisplaced medial femoral neck fractures. The primary outcomes recorded were rates of revision surgery and the incidence of complications.

## 2. Materials and Methods

This study was approved by the local ethic committee and carried out according to the declaration of Helsinki (Ethical commission Number 1335/2016).

For this retrospective cohort study, we examined the data from our clinical database (AKIM, Vienna, Austria) containing details of all patients who were taken to hospital with an undisplaced medial femoral neck fracture between January 2005 and December 2013 (*n* = 453). Details regarding the accident, surgical reports, electronically recorded patient data, and information about the inpatient stay were documented. 

Those who were treated with a DHS (Figure 1) or CSFN (Figure 2) were considered for further analysis.

According to our protocol, patients are encouraged to mobilise for the first 6 weeks with partial weight-bearing, and then progressively increase to full weight bearing by 12 weeks postoperatively. After reaching good mobility status, most patients were transferred to a special nursing facility. Due to a waiting time of about 10–14 days to transfer patients into these special facilities, a lot of the patients were not discharged within this period.

Exclusion criteria included age (<18 years), loss to follow up (e.g., tourists), suffering a pathological, lateral, or displaced medial femoral neck fracture, or a history of osteosynthetic treatment on the same hip (Figure 3). A scientific assistant classified the fractures using the Garden classification based on initial anteroposterior and axial radiographs. Follow-up studies, including x-rays and clinical examinations were carried out for all included patients over a period of at least 24 months.

### 2.1. Outcome Parameters and Complications

The most common complications following osteosynthetic treatment of intracapsular femoral neck fractures, including avascular necrosis (AVN), cut-out, implant movement, refracture, non-union, wound revision (hematoma), postoperative arthrosis, and surgical site infections were recorded in each group. The length of the hospital stay was calculated from the patients’ admission date to their discharge. The duration of the operation itself was extracted from the surgical reports. Furthermore, we analysed the influence of exogenous factors, including tobacco and alcohol consumption, on the development of AVN, which was diagnosed according to the Ficat and Arlet classification on plane radiographs [8]. Early stages of AVN, stage 0 and 1, showed minimal signs on radiographs limiting its diagnosis. However, according to the Ficat and Arlet classification these stages have no impact on treatment. Later stages, identifiable radiographs, were treated with the recommended total or hemi hip arthroplasties.

Patients were stratified based on alcohol consumption, as defined by the recommendations made by the National Health Service of the United Kingdom (NHS). A daily consumption of more than three to four units (or 14 units per week) for men and women was counted as high-risk alcohol consumption [9]. Furthermore, the patients were divided into smokers and non-smokers. Mortality was assessed as part of the outcome, over a two-year period, using data from the Austrian Statistical Office (Statistics Austria, Vienna, Austria).

### 2.2. Statistical Analysis

A Bonferroni correction was applied to compensate for the multiple comparisons carried for the two individual outcomes: incidence of revision surgery and complications. Therefore, the two-sided significance level of 0.05 for nominal variables was reduced to 0.025 for the used χ² test. All other statistical tests were treated as exploratory data analyses and thus not corrected for any multiple testing. The Fisher’s exact test was used due to the small case numbers. Metric variables were examined using an unpaired t-test. A log-rank test was applied to measure the proportion of patients alive after 1, 3, 6, 12, 18, and 24 months. All tests performed were two-tailed. A result was considered significant at a *p*-value smaller than 0.05 using a confidence interval of 95%. Patient data was managed and analysed using SPSS 23.0 statistical software system (SPSS Inc., Chicago, Illinois).

## 3. Results

A total of 453 patients with a medial femoral neck fracture were included in this retrospective cohort study. The mean age of the patients was about 76.9 years (range 18 to 100 years). Baseline demographic data is shown in Table 1.

### 3.1. Rate of Revision Surgery and Complications

A total of 64 (14.1%) cases required revision surgery. No statistical difference was determined between the revision rates among CSFN and DHS patients (13.1% (*n* = 26) vs. 15.0% (*n* = 38); *p* = 0.565, χ² test). Overall, 78 (17.2%) patients suffered from one or more complications during the observation period. Although there is a trend towards fewer complications in the CSFN group compared to the DHS group (13.1% (*n* = 26) vs. 20.5% (*n* = 52) (*p* = 0.038, χ² test), this did not reach the statistical significance threshold for multiple comparisons (α = 0.025). AVN (Figure 4 and Figure 5) was recorded as the most frequent complication within the study population (34 cases; 7.5%). Furthermore, the incidence was almost double among DHS patients (9.4% (*n* = 24) vs. 5.0% (*n* = 10); *p* = 0.076, χ² test). Similarly, the second most common adverse event, cut-out, was observed in 27 patients in total (6.0%), and also doubled in the DHS population (7.5% (*n* = 19) vs. 4.0% (*n* = 8); χ² test *p* = 0.123). Moreover, implant dynamics, meaning a migration of the lag screw(s) in the femoral neck or lateralization of the lag screw(s) due to fracture collapse, were seen in a total of 21 patients (4.6%). No trend was observed among other complications (Table 2).

Those patients who sustained a complication like cut-out, avascular necrosis, refracture or non-union were treated based on their general health state with either hemi hip or total hip arthroplasty. Ten patients developed osteoarthritis in the hip joint. Only three patients were treated with a total hip arthroplasty. The rest of the patients had moderate pain and did not wish further surgical treatment. Those patients with a surgical site infection were treated with local debridement and IV antibiotics. No deep infection with consequent implant removal was observed.

### 3.2. Secondary Outcomes

The mean operative time for the fixation with two cannulated screws was significantly shorter by approximately 13 min (53.35 min vs. 40.77 min, *p* < 0.0001, unpaired t-test). Patients’ length of hospital stay was not significantly decreased when using the CSFN over the DHS (13.2 days vs. 14.2 days respectively, *p* = 0.242, unpaired *t*-test).

Analysis of external factors showed an association between increased alcohol consumption and a significant increase in the incidence of AVN in comparison with people without alcohol consumption respectively (18.6% [*n* = 8] vs. 8.7% [*n* = 41], *p* = 0.035, χ² test) but no significant effect of cigarette smoking (*p* = 0.132, χ² test).

Although 118 (26.0%) patients died within the first 24 months following primary surgery, neither implant significantly impacted survival outcomes (*p* = 0.170, log-rank test). Table 3 shows mortality at each follow up.

## 4. Discussion

The most important finding of the study is that, despite a mean revision rate of 14.1%, current femoral head preserving surgical techniques represent a fairly safe method of treating medial undisplaced femoral neck fractures. This study considered the effects of both DHS and double screw CSFN on the complication and revision rates. So far, there are insufficient studies looking at double screw fixations.

Considering the study’s average revision rate of 14.1%, no significant difference between the two groups could be seen. Based on 28 studies, Slobogean et al. reported a reoperation rate of 18.0% following internal fixation of isolated femoral neck fractures [10]. When equating the CSFN with a triple screw fixation, Jettoo et al. contradicts our conclusion. Their analysis of 52,884 patients concluded that DHS had more favourable outcomes with respect to reducing revision rates [11]. Comparable studies have already shown that treating displaced intracapsular fractures with internal fixation yielded a four-fold increase in revision rates [12].

Although the CSFN was associated with a noticeably lower rate of complications, the correction for multiple comparisons rendered it not statistically significant. Unfortunately, due to different inclusion criteria, complication rates could not be compared with other studies.

It seems reasonable to assume that with fewer complications, revisions will become less frequent. Therefore, it is advisable to recommend primary arthroplasty for patients with poor prognosis or those with displaced fractures of the femoral neck. Comparing internal fixation to hemiarthroplasty, the latter has been established as an advisable choice in recent years [13,14]. In a direct comparison of total hip arthroplasty (THA) and hemiarthroplasty, there was clear evidence supporting the treatment of displaced intracapsular fractures of the femoral neck with THA [14,15]. However, the decision for the optimal surgical procedure should be made in a timely manner, with evidence that salvage THA following a failed internal fixation is associated with a higher risk of complications [16].

The most frequently identified complication in our study population was AVN, which congruently is often listed as the most prevalent complication of hip-sparing management of intracapsular fractures. The high rates of cut-out among this population are likely associated with bone necrosis. Our study described a 5.0% incidence of AVN among those treated with CSFN. This is supported by Krastman et al., who described a very similar technique with the incidence of AVN being 6.0% [17]. Thus, compared to the DHS, the CSFN shows a rarer, but statistically non-significant, occurrence of AVN for the treatment of undisplaced fractures. Applying this observation to the established multiple screw fixation with three screws, Siavashi et al. reported in a randomised clinical trial that there was no difference in the incidence of AVN compared to the dynamic hip screw [18]. A recent paper describing the outcome of 320 patients treated with internal fixation reported an AVN rate of 4.5% for undisplaced and 11.1% for displaced intracapsular fractures with a minimum follow-up of two years [12]. This doubling of incidence further supports that osteosynthetic treatment should only be used in undisplaced femoral neck fractures [10,19,20].

The differences in surgery duration and length of inpatient stay should be highlighted. Smaller incisions and a 13 min faster operating time illustrates the less invasive and more gentle nature of the CSFN approach [21]. Furthermore, the implantation is followed by a briefer hospital stay and is associated with a significantly lower burden on the patients. Cumulatively, these factors represent a considerable economic advantage. Our post-operative data for each surgical method are supported by Jettoo et al., who reported 15- and 13-day post-operative hospitalisation when using a DHS and triple screw fixation, respectively [11]. So far, there is no consensus regarding length of hospitalisation as a potential confounder or predictor of mortality [22,23]. With an approximate 41 min operating time for the implantation of the CSFN, we achieved a comparable surgery time to that by Flóris et al. indicated for the similar Manninger screw (46 min) [21,24].

Not performing a formal cost-benefit analysis, but assuming that an operating theatre costs between €10 and €25 per minute, as previous studies have estimated, the 13-min average extension would equate to an extra cost of €130 to €325 per DHS [25,26]. The extended hospital stay of one day results in an additional €856 in Austria [27]. Taking the €250 higher material costs into account, the final average additional costs are €1236 to €1431 per implanted DHS without considering any follow-up treatment.

Duckworth et al. assumed that alcohol misuse is one of the comorbidities most strongly predictive for fixation failure following a hip fracture in young adults [28]. Likewise, the present study has shown that alcohol consumption increases the rate of AVN. Moreover, we suspect that non-compliance with weight bearing restrictions during rehabilitation, combined with poor bone quality secondary to vitamin D deficiency, play a major disruptive role in fracture healing. Thus, arthroplasty should be recommended as an alternative to internal fixation in such cases [29]. Our search for a link between smoking and the development of AVN has not yielded significant results. Although we failed to show any significant results relating smoking to AVN, there is clear evidence that orthopaedic patients who smoke are more likely to suffer from surgical site infections [30] and more frequently develop non-union in general fracture healing [31].

### Limitations

The present study has several limitations, among which the study design itself is the most important. Its retrospective design did not eliminate a selection bias, allowing the attending surgeon to independently choose the implant. Furthermore, the fractures as classified using the Garden classification showed an uneven distribution between the two groups. In addition, there was no standardisation of the follow-up examinations. Finally, the difficulty in assessing and treating undisplaced medial femoral neck fractures in certain cases requires further prospective randomised trials.

## 5. Conclusions

In conclusion, our data shows a trend supporting the preferential use of the CSFN over the DHS in terms of complication rates but lacks statistical significance. Likewise, it is associated with smaller burden on the patient characterised by a significantly shorter duration of surgery and a briefer length of hospital stay.

## Figures and Tables

**Figure 1 jcm-08-01670-f001:**
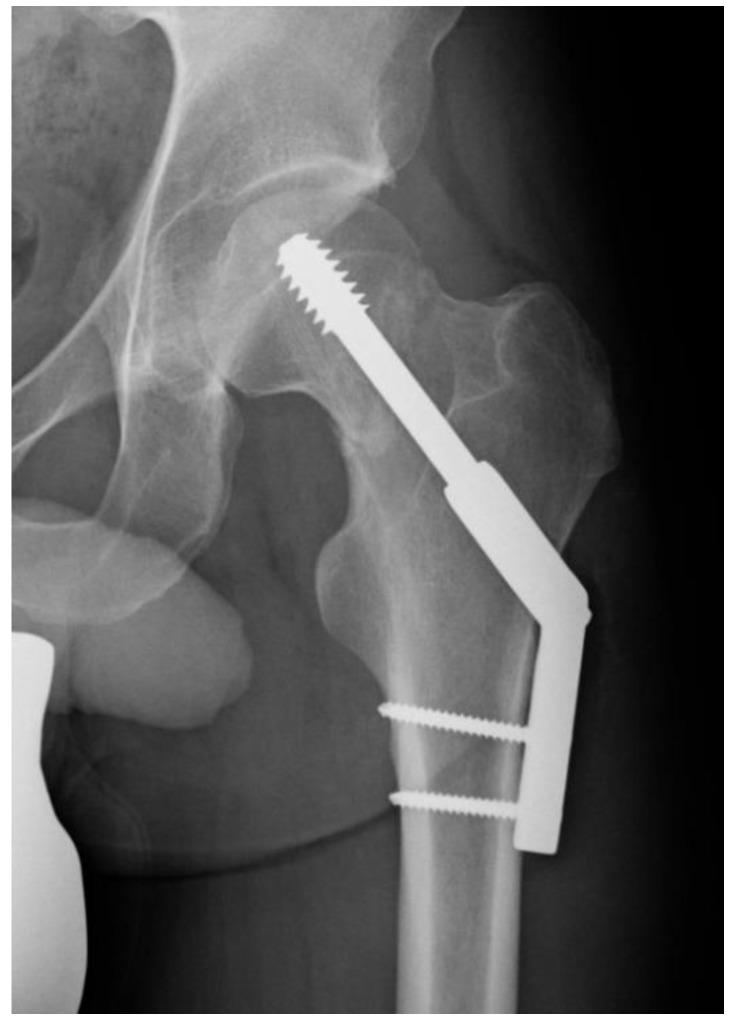
Undisplaced femoral neck fracture after dynamic hip screw (DHS) implantation in anteroposterior view.

**Figure 2 jcm-08-01670-f002:**
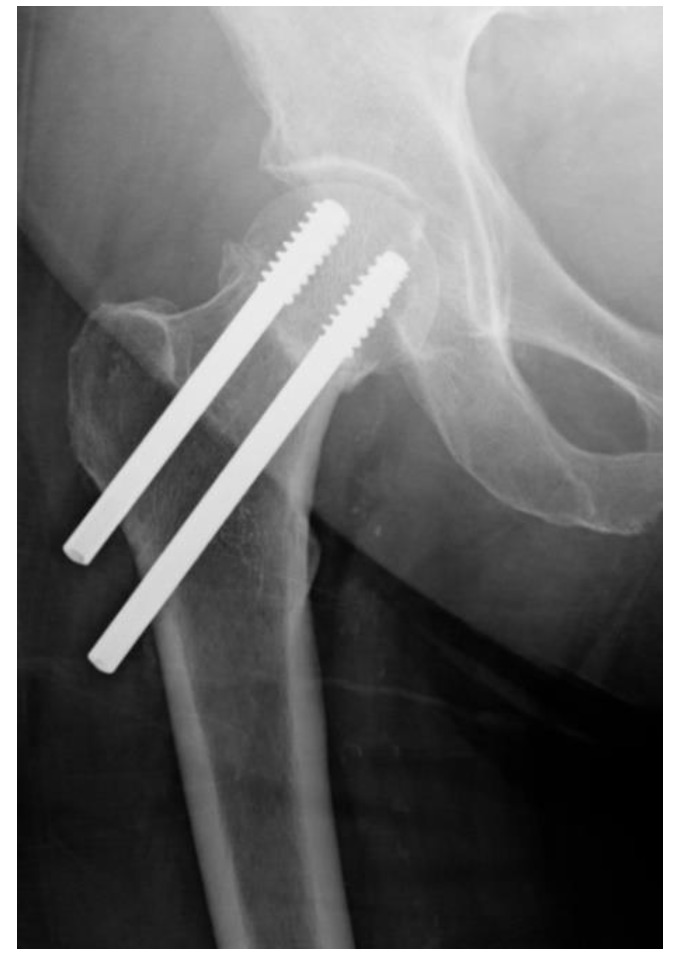
Undisplaced femoral neck fracture after cannulated screw fixation (CSFN) implantation in anteroposterior view.

**Figure 3 jcm-08-01670-f003:**
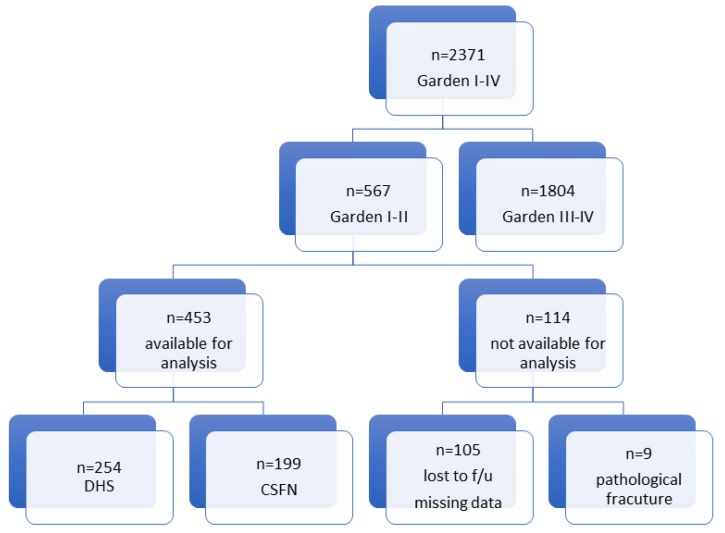
Flowchart showing detailed reasons for exclusion of patients sustaining a medial femoral neck fracture.

**Figure 4 jcm-08-01670-f004:**
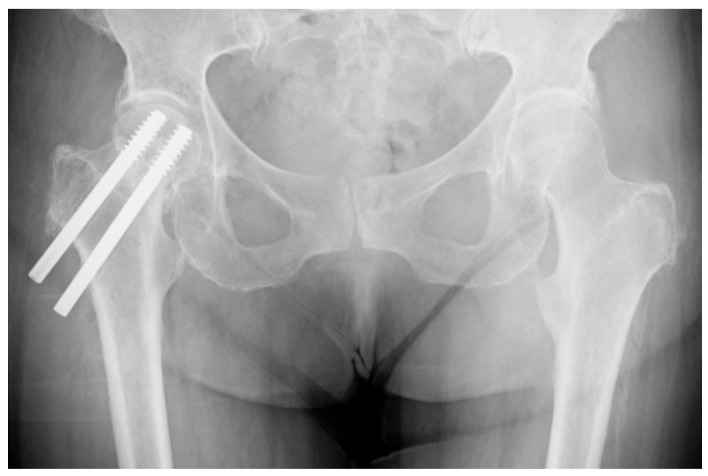
Avascular necrosis development after CSFN implantation in anteroposterior view.

**Figure 5 jcm-08-01670-f005:**
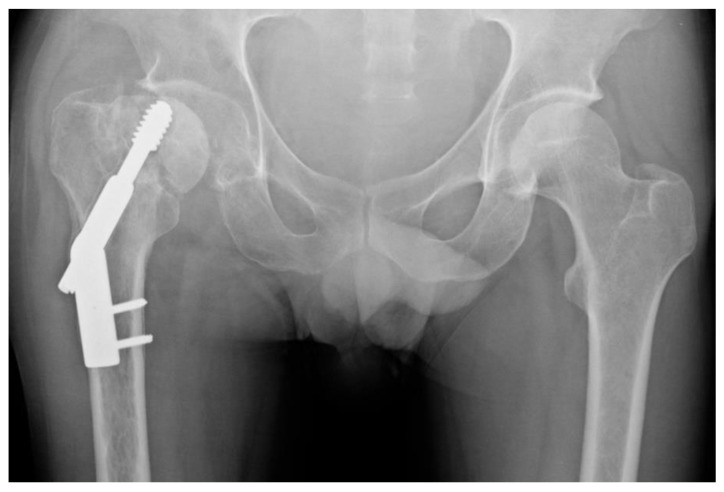
Secondary displacement as a result of avascular necrosis (AVN) after DHS implantation in anteroposterior view.

**Table 1 jcm-08-01670-t001:** Baseline data for all patients included in the study.

	DHS	CSFN
Patients, n (%)	254 (56.1%)	199 (43.9%)
Female, n (%)	178 (51.9%)	165 (48.1%)
Male, n (%)	76 (69.1%)	34 (30.9%)
Mean age, yrs.	76.2	77.9
Mean weight, kg	67.0	64.7
Mean height, cm	168.5	164.5
Mean BMI, kg/m^2^	23.5	23.8

DHS, dynamic hip screw; CSFN, cannulated screw fixation; BMI, body mass index.

**Table 2 jcm-08-01670-t002:** Incidence of post-operative complications.

Type of Complication	DHS	CSFN	Total	*p*-Value
	(*n* = 254)	(*n* = 199)	(*n* = 453)	
Avascular necrosis, n (%)	24 (9.4%)	10 (5.0%)	34 (7.5%)	0.076
Cut-out, n (%)	19 (7.5%)	8 (4.0%)	27 (6.0%)	0.123
Implant dynamics, n (%)	11 (4.3%)	10 (5.0%)	21 (4.6%)	0.727
Refracture, n (%)	4 (1.6%)	1 (0.5%)	5 (1.1%)	0.278
Non-union, n (%)	5 (2.0%)	0 (0.0%)	5 (1.1%)	0.700
Wound revision (hematoma), n (%)	1 (0.4%)	1 (0.5%)	2 (0.4%)	0.862
Postoperative arthrosis, n (%)	6 (2.4%)	4 (2.0%)	10 (2.2%)	0.673
Surgical site infection, n (%)	4 (1.6%)	2 (1.0%)	6 (1.3%)	0.599

DHS, dynamic hip screw; CSFN, cannulated screw fixation.

**Table 3 jcm-08-01670-t003:** Mortality for all patients throughout the 24-month observation period.

	DHS (n = 254)	CSFN (n = 199)
Mortality at 1 month, n (%)	7 (2.8%)	3 (1.5%)
Mortality at 3 months, n (%)	21 (9.4%)	12 (6.0%)
Mortality at 6 months, n (%)	27 (10.6%)	24 (12.1%)
Mortality at 12 months, n (%)	46 (18.1%)	40 (20.1%)
Mortality at 18 months, n (%)	57 (22.4%)	47 (23.6%)
Mortality at 24 months, n (%)	68 (26.8%)	50 (25.1%)

DHS, dynamic hip screw; CSFN, cannulated screw fixation.

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
