# Peer review of "A Comparison of Dynamic Hip Screw and Two Cannulated Screws in the Treatment of Undisplaced Intracapsular Neck Fractures—Two-Year Follow-Up of 453 Patients"

_jcm, 2019, doi:10.3390/jcm8101670_

Round 1
Reviewer 1 Report
Revision was made successfully. I recommend publication.
Reviewer 2 Report
The authors have responded all my comments adequately and made sufficient changes to the manuscript. The manuscript has improved significantly due to the revision.
This manuscript is a resubmission of an earlier submission. The following is a list of the peer review reports and author responses from that submission.
Round 1
Reviewer 1 Report
The paper deals with an important subject of orthopedic trauma, regarding the treatment in intracapsular femoral neck fractures. A retrospective trial was conducted analyzing the data of 453 patients to compare the complication rate in treatment with the DHS versus 2 cannulated screws (CSFN). No significant difference was observed between the revision rates of DHS and CSFN. The complication rate showed slightly an advantage in favor of the CSFN (20% vs 13%). Consumption of alcohol was associated with an increased incidence of avascular necrosis.
In general the manuscript is well-written. However, it has some methological uncertainties to be addressed:
1) How did you define undisplaced medial femoral neck fractures? Garden 1 and 2?
2) You did find initially 453 patients in your patient record with undisplaced fractures. “Those who were treated with a dynamic hip screw or two cannulated screws with a diameter of 9.5 mm without a plate ere considered for further analysis”. How many patients were excluded? A chart about the flow of the patients through your study is essential. How many patients came to the follow up procedure?
3) Femoral neck fractures are the problem in the frail and old population with many comorbidities. Why did you include patients under 70 of age? Introduction focuses on osteoporotic fractures as well.
4) The common procedure in medial femoral neck fractures is the additional implantation of an antirotational screw. Excessive sintering and cut-out depend on rotation. Rotational stability is very important in treatment and anchorage mechanism as well. Please discuss shortly the biomechanical advantages of the implant designs with rotational stability in the introduction section, etc., (see Knobe M et al. Screw-blade fixation systems in Pauwels three femoral neck fractures: a biomechanical evaluation. Int Orthop. 2018 Feb;42(2):409-418.)
5) An important issue is function and quality of life in the geriatric population. Assessing function using scores would improve the quality of your manuscript.
6) Complications: What about Tractus irritation in CSFN? The screws seem very long and when fracture sintering (with femoral neck shortening) occur, this problem should be evident.
7) AVN and cutout more frequent in DHS?! The problem is the femoral neck shortening resulting in fracture collapse and failure. How did you underline your diagnosis AVN? See failure mechanisms in femoral neck fractures: Knobe M et al. Screw-blade fixation systems in Pauwels three femoral neck fractures: a biomechanical evaluation. Int Orthop. 2018 Feb;42(2):409-418.)
8) What do you mean with Implant dynamics? Please specify
9) To determine the failure mechanism a quantification of the femoral neck shortening radiographically is necessary.
10) Among your outcome parameter one big issue is missing: the quality of reduction and screw placement. The most complications are due an in adequate surgical treatment.TAD and quality of reduction are necessary.
11) Patients’ length of hospital stay seems to me very long (13/14 days). Please clarify.
12) Revision rate of 14.1%: Please specify the revision procedures in detail.
13) Postoperative management: Please give an explanation for your protocol. Full-weight bearing?
14) How did you assess the influencing factors on complications and mortality or AVN? Did you use a multivariate analysis? What about complications due to comorbidity? Cardiovascular complications? See: Carow J et al. Mortality and cardiorespiratory complications in trochanteric femoral fractures: a ten year retrospective analysis. Int Orthop. 2017 Nov;41(11):2371-2380.
Reviewer 2 Report
In this retrospective study, the authors compare the outcomes of DHS and CSFN treatment of undisplaced intracapsular femoral neck fractures. The results have a trend towards favoring the CSFN treatment. The results are interesting and may be of importance when designing optimal surgeries. Overall, the manuscript is well-written. However, there are a number of issues to be addressed before consideration for publication.
1) The manuscript needs to be proof-read to fix grammar and working issues and for the structure and presentation. Some examples (not limited to these):
Page 1, line 38. "... age-associated bone fractures also present more frequently.." Is there a verb missing?
Page 1, lines 38-39. "The principle factors responsible for this are primarily the age-related bone demineralization..." is written in plural although only 'deminalization' is presented as the factor
Page 3, lines 77-79. THis is an aim paragraph that should be compiled with the aim presented in Introduction.
Page 3, lines 91-92. "who drank below the NHS recommended amount". Sound like NHS recommends drinking.
Page 4, table 1. Typo in mean height CSFN
Page 5, table 2 caption. State clearly in the aims the primary and the secondary aim using these wordings to make it more reader-friendly
2) Samples were excluded due to numerous reasons. What was the number of excluded and included patients in the end? Please add details.
3) Page 2, lines 69-70. "Follow-up studies were carried out for all patients ..." For all or for included patients?
4) Not all readers are familiar with terms like 'avascular necrosis' or 'cut-out %'. Especially, how these are evaluated from radiographs is not part of everyones knowledge. Please add short descriptions of AVN and cut-out and how are these observed from the radiographs.
5) The results would be easier to read if together with the percentages, also number of patients would be presented.
6) Page 5, line 133, "...AVN (18.6% [n = 8] vs 8.7 % [n = 41],..." what are numbers, please be more specific.
7) Do the authors have the data on the surgeons who performed each operation? It is discussed in the limitations that the selection bias of the surgeons of what operation to perform could not be eliminated, but could this information be added to the statistics? Maybe the surgeons who choose to use DHS operate slower than those who choose CSFN and the time difference between the surgeries is because of this, or even vice versa and the time difference could be even larger?
8) Be consistent in using %-signs within parenthesis in tables 1,2, and 3. 1 and 3 do not have them, 2 has.